# Occurrence of *Chlamydia* spp. in Conjunctival Samples of Stray Cats in Timișoara Municipality, Western Romania

**DOI:** 10.3390/microorganisms10112187

**Published:** 2022-11-04

**Authors:** Andreea Tîrziu, Viorel Herman, Kálmán Imre, Diana Maria Degi, Marius Boldea, Vlad Florin, Timea Andrea Bochiș, Marcu Adela, János Degi

**Affiliations:** 1Ophthalmology Department, “Victor Babes” University of Medicine and Pharmacy, Piața Eftimiu Murgu No. 2, 300041 Timisoara, Romania; 2Department of Infectious Diseases and Preventive Medicine, Faculty of Veterinary Medicine, University of Life Science “King Michael I” from Timișoara, 300645 Timisoara, Romania; 3Department of Public Health, Faculty of Veterinary Medicine, University of Life Science “King Michael I” from Timișoara, 300645 Timisoara, Romania; 4Department of Pharmacology and Pharmacy, Faculty of Veterinary Medicine, University of Life Science “King Michael I” from Timișoara, 300645 Timisoara, Romania; 5Department of Exact Sciences, Faculty of Agriculture, University of Life Science “King Michael I” from Timișoara, 300645 Timisoara, Romania; 6Department of Semiology—Preclinical I, Faculty of Veterinary Medicine, University of Life Science “King Michael I” from Timișoara, 300645 Timisoara, Romania; 7Department of Animal Production Engineering, Faculty of Bioengineering of Animal Recourses, University of Life Science “King Michael I” from Timișoara, 300645 Timisoara, Romania

**Keywords:** *Chlamydia* spp., conjunctivitis, stray cats, zoonosis

## Abstract

Despite the widespread public health concern about stray cats serving as reservoirs for zoonotic agents, little is known about the effect of urban and peri-urban landscapes on exposure risk. We conducted this study to monitor the presence of *Chlamydia* spp. in stray cats, with or without conjunctivitis, living in Timișoara Municipality, Western Romania, using staining and PCR methods. A total of 95 cats were enrolled, and conjunctival samples were harvested from 68 clinically healthy cats and another 27 cats presenting with clinical signs of conjunctivitis. Overall, we found that 65.3% (62/95) of the cats tested positive for *Chlamydia* spp. by PCR. *Chlamydia* spp. were detected in 45/95 conjunctival samples using a standard Giemsa stain, compared with 62/95 using PCR (Cohen’s kappa index = 0.308; *p* = 0.0640). Of the cats that tested positive by PCR, 72.6% (45/62) were asymptomatic, and another 27.4% (17/62) expressed clinical signs of conjunctivitis. We found no significant difference between (*p* > 0.05) the distribution of infection and the recorded epidemiological data (sex, breed, age, territorial distribution, or sampling season). However, the *Chlamydia* spp. detection frequency was significantly higher in asymptomatic than in symptomatic cats (*p* = 0.0383). The obtained results increase the level of concern and awareness about the possible zoonotic potential of this pathogen and highlight that urban stray cats can be essential sources of feline chlamydiosis.

## 1. Introduction

The domestic cat (*Felis catus*) is one of the most abundant carnivores found in densely populated urban and peri-urban areas worldwide [1]. Feral cats are the offspring of domestic cats that have adapted to living on their own in rural and urban areas, but they also include unowned domestic cats that have left or lost their home and are surviving in the urban environment as stray cats [2,3].

Due to their behavior, both types of cats are referred to as free-roaming cats. As over 75% of the emerging human infectious diseases are zoonotic [4,5,6,7], more information on the pathogens that can be transmitted to sympatric wildlife or humans is needed. In this context, free-roaming cats (including stray cats) can pose a considerable public health risk since they live as large populations in close proximity to humans and can harbor pathogens that cause infections in humans and animals [8]. The transmitted pathogens are primarily determined by host communities, with the presence and frequency of free-roaming pets remarkably impacting the risk of disease transmission to humans. Viruses, bacteria, and parasites are typically prevalent in free-roaming cats due to the increased infection risk from outdoor access [9,10,11] and their roaming behavior [12]. Free-roaming cats have unrestricted access to public spaces, such as parks and playgrounds, which can be populated by various non-human hosts capable of transmitting different pathogens [13,14].

*Chlamydia* spp. are intracellular pathogen with zoonotic potential that can spread between humans and animals, including birds [15], and infects humans [5,16]. Depending on the host and chlamydial species, chlamydiosis in animals can range from infection with no symptoms to severe disease that can kill the animal [5,17]. The symptoms of chlamydial infections include mild to severe conjunctivitis; rhinitis; pneumonia; mastitis; arthritis/polyarthritis; pericarditis; polyserositis; encephalomyelitis; placentitis, which causes abortion, stillbirth, or weak newborns; endometritis/metritis; orchitis/epididymitis/urethritis; infertility; and enteritis [17,18]. Certain *Chlamydia* species are contagious and pose a considerable public health risk because they can cause pneumonia, atherosclerosis, and coronary heart disease, among other serious complications [5,19].

The genus *Chlamydia* includes a sexually transmitted disease agent that is also the causative species for human conjunctivitis, *Chlamydia trachomatis*; a mouse pneumonia agent, *Chlamydia muridarum*; and a pig pneumonia agent, *Chlamydia suis*. Other *Chlamydia* spp. include a human pneumonia agent, *Chlamydia pneumoniae*; and the animal disease or zoonosis agents *Chlamydia psittaci*, *Chlamydia caviae*, *Chlamydia pecorum*, *Chlamydia abortus*, and *Chlamydia felis*, which infect birds, hamsters, cows, sheep, and cats, respectively [5,17,20]. Likewise, *C. felis* can cause various illnesses in non-human mammals and birds, including atypical pneumonia, enteritis, conjunctivitis, endocarditis, and abortion, resulting in considerable financial losses in related industries [17,18,21,22]. *C. felis* is a primary and remarkable cause of feline conjunctivitis and can serve as a nasal epithelium pathogen [15,20,23,24]. Conjunctivitis typically develops in infected cats after an incubation period of 2–7-days [25]. In most cases, conjunctivitis caused by *C. felis* is unilateral, but cases with bilateral eye damage have also been described [26]. Some complications can be encountered during the disease course, including conjunctival chemosis, blepharospasm, ocular discharge, and nictitating membrane hyperemia [23,25]. Initially, conjunctival discharge is serous and can become mucoid to mucopurulent [25]. Additional signs may include fever, lethargy, enlargement of the submandibular lymph nodes, lameness, inappetence, and decreased weight gain (in kittens) [23].

In cats, *Chlamydia* spp. primarily target the conjunctival epithelium and cause acute to chronic or recurrent conjunctivitis, particularly in young cats (from 2 to 12 months) [27]. Infected cats have mucoid or mucopurulent to follicular conjunctivitis [17,27]. Chlamydial inclusions can be demonstrated in Giemsa-stained smears prepared from conjunctival scrapings, but PCR-based methods are more sensitive and specific [28].

In the absence of any published scientific report in Romania about the presence of *Chlamydia* spp. in any potential host, in the present study, we aimed to investigate the detection rate of this pathogen in stray cats living in urban and peri-urban areas of the Timisoara Municipality, Western Romania, using conventional and molecular tools.

## 2. Materials and Methods

### 2.1. Sample Collection

Ninety-five conjunctival swab samples were collected from stray cats located in different areas of the Timisoara Municipality, Western Romania, between June 2020 and March 2021. Specimen collection was carried out in accordance with the CLSI document M29-A4 [29] and in agreement with the regulations imposed by the Romanian Veterinary College (protocol numbers 34/1.12.2012) and the current practices of the University Veterinary Clinics of the Faculty of Veterinary Medicine from Timisoara, Romania. Samples were obtained from anesthetized cats housed as part of the stray cat trap–neuter–return program organized by the Network for Animal Protection (NetAP) health organization.

A veterinarian, who was a specialized ophthalmologist, assessed the presence (*n* = 27) or absence (*n* = 68) of conjunctivitis and checked for ocular serous, mucous or mucopurulent discharge; blepharospasm; conjunctival hyperemia; and chemosis as signs of conjunctivitis in the monitored cats. Before sampling, the animal eyes were cleaned with a cotton swab to remove mucosal discharge. Conjunctival samples were collected from the upper tarsal conjunctiva using a Dacron polyester-tipped swab (Thermo Fisher Scientific, Waltham, MA, USA), consisting of one swab from each eye. We obtained the specimens by reflecting the eyelid and firmly passing the swab across the conjunctiva four times, with a quarter turn between each pass. Subsequently, we thinly spread the conjunctival scrapings on a single sterile glass slide. The slides were left to air-dry for 15 min to 6 h before being fixed for 5 min with absolute methyl alcohol (Chemos GmbH & Co. KG, Altdorf, Germany) at room temperature. They were refixed in methanol before stain for 15–20 min with May–-Grünwald–-Giemsa stain (Merck KGaA, Darmstadt, Germany). To obtain a bluer coloration, the slides were immersed in a working phosphate buffer (Merck KGaA, Darmstadt, Germany) (20–70 mmol/L) at pH 7.2 for 1.5 min and then briefly rinsed in deionized water (Merck KGaA, Darmstadt, Germany). We evaluated the slides after air drying.

The colored smears were microscopically examined using the same reader. At least thirty fields were examined using a 25× objective (200× total magnification), and then another thirty fields were examined using an oil immersion lens, usually 100× (800× total magnification).

All the swab samples were stored at −20 °C until further analysis. During sampling, we recorded several epidemiological data, including age (according to their completion of the primary dentition), sex, clinical signs of conjunctivitis, and sampling data.

### 2.2. Microscopy Assay and PCR Test

First, the May–Grünwald–Giemsa staining method was used to determine the cellular composition of the harvested ocular samples. *Chlamydia* spp. are basophilic organisms with bright magenta-stained chlamydial inclusions ranging from round to oval in epithelial cells [27,30].

Subsequently, we screened all of the conjunctival samples by PCR for molecular investigation. PCR was designed to detect the chlamydial outer membrane protein gene (~270 base pairs) using the cycling conditions previously described by Buxton [31] and the oligo 420 (5′-CAG GAC ATC TTG TCT GGC TTT AA-39) and oligo 423 (5′-CGG ATG CTG ATA GCA TCA CAC CAA GT-39) primer sets. Within each PCR reaction, we used a positive control consisting of the *Chlamydia* spp. vaccine DNA (Purevax RCPCh, Merial, France) and PCR-grade distilled water as a negative control. Bacterial DNA was extracted from the resuspended swab specimens in 1 mL of a medical-grade saline solution. The initial working volume consisted of 200 µL of the sample, and we used a PureLink™ Genomic DNA mini kit (Invitrogen™, Carlsbad, CA, USA) for genomic DNA isolation following the manufacturer’s instructions. The final amplification products (20 µL) of each reaction were analyzed by electrophoresis on a 2% agarose gel, stained with ethidium bromide, and subsequently visualized using an image capture system under ultraviolet illumination.

### 2.3. Statistical Analysis

We performed statistical analysis of the prevalence of *Chlamydia* spp. infection in relation to the recorded epidemiological data (sex, breed, age, sampling season, and clinical signs of conjunctivitis) based on Pearson’s chi-square (*χ2*) test (Microsoft Excel 2007, Redmond, WA, USA). *p* ≤ 0.05 was considered to indicate a statistically significant difference.

## 3. Results and Discussion

Overall, the smear results for 45 (47.4%) of 95 specimens’ revealed typical chlamydial inclusions in the May-Grünwald–Giemsa-stained epithelial cells. Inclusions appeared as discrete masses in the cytoplasm consisting of particles ranging from the small (~300 nm) red to purple staining elementary bodies to the larger (1 µm) dark blue staining initial bodies (Appendix A—microscope images). The elementary body inclusions were usually large and contained many particles. The initial body inclusions were smaller and contained fewer particles. Not infrequently, we found these two types of particles in the same inclusion.

We found that inclusion of positive samples. Other cytological findings occurred less frequently and in samples with fewer inclusion-positive smears.

Inclusions were absent in 50 conjunctival samples (50/95; 52.6%), which contained only degenerated neutrophils and an abundance of coccus bacteria cells. In the distribution of the *Chlamydia* positive findings in of the cats that tested positive by PCR, 72.6% (45/62) were asymptomatic, and another 27.4% (17/62) expressed clinical signs of conjunctivitis he May–Grünwald–Giemsa-stained smears according to the expressed sign of conjunctivitis was 64.70% (11/17) in stray cats with conjunctivitis and 75.6% (34/45) in asymptomatic stray cats.

We found that 62 stray cats (62/95; 65.2%) with clinically suspected feline chlamydiosis conjunctivitis were *Chlamydia* spp. positive by PCR. During the clinical examinations, the cats expressed signs of conjunctivitis, including redness and swelling of the conjunctival tissue, with or without ocular discharge. Notably, we observed keratitis characterized by corneal opacity in 17 cats and an entire eye globe rupture accompanied by purulent discharge accumulation in one other.

### Chlamydia spp. Detected in Stray Cats

Table 1 summarizes the results of *Chlamydiaceae* family detection using May–Grünwald–Giemsa staining and the specific PCR method. Conjunctivitis was diagnosed in 28.5% (27/95) of the investigated stray cats, and another 71.5% (68/95) exhibited no symptoms during sampling and were classified as asymptomatic. We estimated that 82.1% (78/95) of the animals demonstrated closeness and friendship with the persons who participated in their collection.

Out of the 95 total conjunctival samples, 17/27 (62.9%) stray cats with suspected follicular conjunctivitis and 28/68 (41.1%) asymptomatic stray cats tested positive for *Chlamydia* spp. according to the Giemsa staining method. A more detailed depiction of the results, which is based on the recorded epidemiological data, is demonstrated in Table 1.

We revealed that when using the PCR technique, 65.3% (17/27, or 62.96% symptomatic and 45/68, or 66.17% asymptomatic) of the stray cats in the Timisoara Municipality, Romania, were positive for *Chlamydia* spp.

We performed PCR for *Chlamydia* spp. detection using the primers corresponding to the conserved regions in the upstream noncoding region and 5’ coding region of the gene for the chlamydial major outer membrane protein, which encodes an outer membrane protein that is thought to be an important antigen targeted by the host immune system in protection [32].

Even if the registered overall prevalence obtained during molecular investigations was higher than the prevalence obtained using the staining technique (65.2% vs. 47.3%), the results must be interpreted with caution, as inhibition is a common phenomenon during diagnostic PCR [33]. The occurrence of PCR-negative results in some microscopically positive *Chlamydia* samples can be explained by the possible presence of PCR inhibitor substances in the tested samples, as has been previously suggested by other authors [33].

The significant association between PCR detection and *Chlamydia* spp. conjunctivitis indicates that the presence of the pathogen is associated with the disease. PCR is typically used to detect specific *Chlamydia* spp. target genes in the conjunctiva [28]. Consequently, implementing highly sensitive methods may answer questions regarding the relatively high prevalence of *Chlamydia* spp. infection [17]. Detecting *Chlamydia* spp. infection is of utmost importance because this pathogen is highly contagious, and many stray cats live in the study area.

We found a fair agreement between the May–Grünwald–Giemsa staining and specific PCR methods for detecting *Chlamydia* spp. (Cohen’s kappa index = 0.308) in the monitored stray cats. In this regard, some of the microscopically *Chlamydia*-negative (*n* = 50) May–Grünwald–Giemsa-stained samples were PCR-positive (*n* = 62) and vice versa (33 May–Grünwald–Giemsa-stained positive samples vs. 45 PCR-negative samples).

The results show a high proportion of *Chlamydia* spp. in the etiology of conjunctiva infections and an endemic presence of this pathogen in the stray cat population.

The stray cats came from different zones of the Timisoara Municipality that share similarities in terms of living conditions, with environments typical of densely populated, multi-generational, and impoverished habitats.

According to the results of several field studies conducted in various countries (using PCR, isolation in cell cultures, Giemsa and Diff Quick stain, or immunofluorescence assays), the prevalence of chlamydial infection in pet cats ranges between 0 and 10% in healthy animals [34] and 5.6–30.9% in cats with conjunctivitis [34,35,36,37,38]. The prevalence is typically higher in stray cat populations, ranging from 24.4 to 35.7% overall [38,39,40,41], but it can reach 65.8% in subgroups with conjunctivitis [40].

The differences in the detected prevalence of *Chlamydia* spp in cats can be associated with some factors, including geographical and ecological factors, clinical illness status, age of cats, territorial distribution, and employed diagnosis techniques. *Chlamydia* spp. prevalence rates are usually higher in the summer months [17,20,27].

We consider this study to be a promising first step toward assessing the health status of stray cats in Romania and gaining information on possible zoonotic hazards.

In addition, we previously reported that 5.0% of small-animal-clinic veterinarians were seropositive for *C. felis* [35]. These results raise the possibility that *Chlamydia* spp. are zoonotic [5,16].

In the results of this study, 65.3% of all the conjunctival samples taken from urban stray cats in the Timisoara Municipality, Romania, were PCR-positive for *Chlamydia* spp., but the prevalence varied between metropolitan areas. These differences can be attributed to population density variations and stray cats’ nutritional and overall health status [34,42,43]. Considerable variations in positivity rates between stray cat populations can be found in the results of studies from other countries, ranging from around 5% in healthy cats to 23.3–65.8% in symptomatic animals [40,44]. This is consistent with our current findings, which indicate that 62.9% (17/27) of the symptomatic and 66.1% (45/68) of the asymptomatic urban (Timisoara Municipality) stray cats tested positive for *Chlamydia* spp. by PCR, compared with the less-sensitive Giemsa staining method (*p* = 00640).

The fair agreement (k index = 0.308) observed between the screening methods suggests that in order to obtain a clearer evaluation of the *Chlamydia* spp., a combination of staining and PCR methods should be used when investigating the cats’ infective status. While we detected a fair agreement (k index = 0.308) between the two screening methods used (PCR, staining method), the occurrence of PCR-positive/Giemsa-negative (*n* = 28) and PCR-negative/Giemsa-positive (*n* = 11) samples suggests that the combined use of both screening methods is advantageous to ensure effective evaluation of the infection status of cats.

When comparing chlamydia infection occurrence according to sex, we found that of the 58 samples collected from stray male cats (42 asymptomatic and 16 with conjunctivitis), 49/95 were positive in at least one of the tests (PCR and Giemsa stain), corresponding to 51.6% of the stray male cats. Of the 37 conjunctival samples collected from female cats (26 asymptomatic and 11 with conjunctivitis), 24/95, or 25.3%, were positive in at least one of the PCR analyses and Giemsa stain methods (Table 1).

By comparing the relative risk of chlamydial infection across the stray cat breeds, we found that the European stray cats had a higher chlamydial infection risk than the non-European breeds of stray cats (*p* = 08776). The positivity rate for *Chlamydia* spp. in the European cats was 61.0% (58/95) and 14.7% (14/95) in the non-European stray cats (Table 1).

We also observed noticeable differences in *Chlamydia* spp. prevalence between stray cats >2 and <6 months old and >6 months and <2 years old in the urban region of the Timisoara Municipality, Romania (*p* = 0.2864). Regarding the age when the stray cats were most likely to be positive for *Chlamydia* spp., the >2 months and <6 months age (37/95; 39.0%) had the highest positivity rate (Table 1). The association between age and the occurrence of *Chlamydia* spp. infection is most commonly reported for cats younger than one year old [23,40].

During the warm season, we sampled most of the stray cats in June and September (61/95), and we investigated the impact of the sampling season and distribution on chlamydial positivity rates. When comparing *Chlamydia* spp. prevalence in stray cats with conjunctivitis and without clinical signs, 51.6% (49/95) of the stray cats were confirmed positive in the warm sampling season compared with 25.3% (24/95) of the cats sampled during the cold season (*p* = 0.3324).

We more frequently detected *Chlamydia* spp. in asymptomatic cats (57.9%; 55/95). When comparing the relative risks of *Chlamydia* spp. between stray cats with clinical signs of the disease and stray cats without clinical symptoms of conjunctivitis, we discovered that *Chlamydia* spp. risk in cats without clinical signs of infection was higher than in stray cats with clinical signs of conjunctivitis (*p* = 00383).

We found that conjunctivitis was the most common clinical sign of chlamydial infection, with the prevalence of purulent discharge, chemosis, hyperemia, blepharospasm, and mucopurulent discharge. However, not all the infected cats exhibited conjunctivitis symptoms; consequently, the absence of clinical signs does not rule out infection.

Feline chlamydiosis is one of the most common infections in cats [23], but no information is available regarding its infection prevalence in Romania. In the current study, the results show that 62.96% prevalence of *Chlamydia* spp. in stray cats with clinical signs of conjunctivitis. Because studies that focus on screening to detect the most common feline conjunctiva pathogens in Romania are lacking, the proportion that *Chlamydia* spp. pathogens represent of total cat pathogen infections remains unclear.

Direct contact is required for *Chlamydia* spp. transmission because of its poor environmental persistence, and certain environmental risk factors, such as outdoor living in a multi-cat environment, increase the amount of contact between cats [34,45]. The fact that stray cats are more likely to contract *Chlamydia* spp. than domestic cats may be because they are more likely to freely roam and share a habitat with other animals or to be fed in highly trafficked areas. As discussed earlier, *Chlamydia* spp. infection is most commonly reported for cats younger than one year old [23,40].

According to a previous study, poor environmental conditions are associated with a high rate of feline chlamydiosis [37]. The finding that *C. felis* is present in stray cats living in a multi-cat environment supports previous evidence of a higher chlamydiosis prevalence in cats living in a multi-cat environment and in younger and stray cats [34]. In areas where stray cats are fed and thus gather in groups, the prevalence of pathogens that are transmitted through direct contact is higher [43,44,45]. In this study, we also detected *Chlamydia* spp. in an older cat in a severe and fatal condition. We assumed that chlamydiosis was not the only disease that the cat had, given the possibility of co-infections (including FIV, FeLV, and *Haemobartonella felis* infections). Additionally, uncommon chlamydiosis manifestations were apparent, including corneal inflammation involvement and complete eye-globe rupture [27].

Because the morphological characteristics of *Chlamydia* spp. are readily identifiable, a routine cytological examination is a reliable diagnostic tool for its detection [46,47]. However, *Chlamydia* spp. are rarely cytologically identified in subclinical or chronic cases [27]. Because of their exceptional sensitivity and specificity, a molecular-level assay is recommended for use in clinically suspected *Chlamydia* spp. patients [45,48,49,50].

The *Chlamydia* spp. pathogen can cause severe infection and gradually affect the host’s immune system [17]. Thus, the early and accurate detection of chlamydiosis may aid in preventing the spread of this potentially fatal infection [27]. The main results of our survey indicate the emergence of *Chlamydia* spp. infection in cats living in high-population areas of the Timisoara Municipality, where the epidemiological situation of this disease was previously completely unknown. Without proper treatment, the infection can become fatal and represents a burden [51]. The findings of this study heighten our awareness of the potential severity of *Chlamydia* spp. infections, as well as concern for human health as a result of the zoonotic potential of *Chlamydia* spp. [13,19,20,45,52], particularly among people feeding stray cats and those living in the surrounding environment.

Seroprevalence for *C. felis* is relatively high in many countries, including China, Italy, Japan, and Slovakia, particularly among stray (>10%) and house cats (>3%) [15,34,53,54,55]. Although cats mainly carry the bacterium, dogs are also potential reservoirs of *Chlamydia* spp. [15]. Therefore, the ubiquity of cats and their interactions with humans may facilitate *C. felis* dissemination to humans [15,34,54,55]. A previous seroepidemiological study in Japan showed that regarding prevalence, 1.7% of the general population and 8.8% of small animal clinic veterinarians had antibodies against *C. felis* proteins [53].

It is possible that the recorded higher prevalence of chlamydial infections in stray cats is due to the fact that they are fed in high-traffic areas and are free to roam, with both situations increasing the likelihood of their exposure to other animals. However, chlamydial infection incidences in cats are usually higher in the spring and summer [15,34].

## 4. Conclusions

In the results of the current survey, the occurrence of *Chlamydia* spp. infection in stray cats has been demonstrated for the first time in Romania, with a considerably higher detection frequency in asymptomatic than in symptomatic cats. This study provides valuable information for veterinarians in managing cat ocular infections, suggesting that a combination of staining and PCR techniques can increase the accuracy of the diagnosis of the disease. However, the obtained results suggest that for the confirmation of *Chlamydia* infection, the PCR technique is the best option, being recommended also in clarifying doubts during staining. Furthermore, this study opens the opportunity to initiate new investigations addressing some limitations of the current findings, especially regarding the evaluation of the zoonotic potential of isolates using species-specific PCR tools. The *Chlamydia* species that is the causative agent of feline chlamydiosis is yet to be confirmed, although *C. felis* is the most likely candidate.

## Figures and Tables

**Table 1 microorganisms-10-02187-t001:** Distribution of *Chlamydia* spp. infections in the monitored stray cats according to the recorded epidemiological data.

Recorded Epidemiological Data	Asymptomatic Stray Cats (*n* = 68)			Symptomatic Stray Cats (*n* = 27)			Total Positivity—PCR + Giemsa (%)	*p*-Value
	No. Samples Collected	Positive PCR	Positive Giemsa Stain	No. Samples Collected	Positive PCR	Positive Giemsa Stain		
**Sex**								*p* = 0.024
Male	42	27 (64.2%)	19 (45.2%)	16	11 (68.7%)	10 (62.5%)	49 (51.6%)	
Female	26	18 (69.2%)	9 (34.6%)	11	6 (54.5%)	7 (63.63%)	24 (25.3%)	
**Breed**								*p* = 0.8776
European	51	25 (49.0%)	18 (35.3%)	23	19 (82.6%)	20 (86.9%)	58 (61.0%)	
Non-European	17	14 (82.3%)	5 (29.4%)	4	4 (100%)	2 (50.0%)	14 (14.7%)	
**Age (approximately)**								
˂2 months	3	1 (33.3%)	-	2	1 (50.0%)	1 (50.0%)	3 (3.2%)	*p* = 0.2864
>2 months and <6 months	32	27 (84.3%)	21 (65.6%)	10	8 (80.0%)	6 (60.0%)	37 (39.0%)	
>6 months and <2 years	22	14 (63.6%)	11 (50.0%)	9	4 (44.4%)	3 (33.3%)	20 (21.0%)	*p* = 0.0919
>2 years	11	3 (27.2%)	1 (33.3%)	6	4 (66.6%)	2 (33.3%)	13 (13.7%)	*p* = 0.1248
**Sampling season**								*p* = 0.3324
Warm	48	32 (66.6%)	24 (50.0%)	13	9 (69.2%)	3 (33.3%)	49 (51.6%)	
Cold	20	13 (65.0%)	9 (45.0%)	14	8 (57.1%)	3 (37.5%)	24 (25.3%)	
**Clinical sign of conjunctivitis**								*p* = 0.0383
Yes	27	17 (62.9%)	17 (62.9%)	-	-	-	18 (19.0%)	
No	68	45 (66.1%)	28 (41.1%)	-	-	-	55 (57.9%)	
**Metropolitan area**								
Western	43	31 (72.1%)	27 (62.8%)	15	9 (60.0%)	4 (26.67%)	38 (40.0%)	*p* = 0.1877
Eastern	25	11 (44.0%)	7 (28.0%)	12	7 (58.3%)	5 (41.7%)	22 (23.16%)	

## Data Availability

Not applicable.

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
