# Peer review of "Occurrence of Chlamydia spp. in Conjunctival Samples of Stray Cats in Timișoara Municipality, Western Romania"

_microorganisms, 2022, doi:10.3390/microorganisms10112187_

Round 1

Reviewer 1 Report (Previous Reviewer 4)

My comments in attached file.

Author Response

Response Reviewer 1 (Revision 1)

Query (Q) 1: Authors did not explain why inhibition control was not used in PCR (it was in my previous comments).

Answer: The authors honestly acknowledge the fact that it would be helpful to the using of PCR inhibition control during investigations. However, this limitation has been highlighted within lines 187-191 of the previously revised version "Even if the registered overall prevalence obtained during molecular investigations was higher than the prevalence obtained using the staining technique (65.2% vs. 47.3%), the results must be interpreted with caution, as the inhibition is a common phenomenon during diagnostic PCR [33]. The occurrence of PCR negative results in some microscopically positive Chlamydia samples can be explained by the possible presence of PCR inhibitor substances in the tested samples, as has been previously suggested by other authors [33]." According to the author’s knowledge, presently, there is no consensus on how to detect it in medical microbiology routine practice investigations, and it is not recommended as mandatory action. In addition, the financial support for the reagents of the present study was very limited. Special thanks for your understanding!

Q2: Taking to account the avilabilty of methods for identyfication different Chlamydia species it is not good that Authors did not perform this identyfication. It is very important what kind of Chlamydia species was present in samples from cats. If the authors did not perform species identyfication they should add information about necessity species identification in the future studies. Generally, lack of species identification causes that value of the manuscript is lower.

Answer: During previous revision, the authors acknowledge the fact that the using of species-specific PCR would increase the study value. However, this limitation was highlighted in the conclusion section: Furthermore, this study opens the opportunity to initiate new investigations addressing some limitations of the current findings, especially regarding evaluation of the zoonotic potential of isolates using species specific PCR tools (in the revised manuscript version, lines 311-313).

Q3: Line 118: 2.2. I suggest change the title on: Microscopy assay and PCR test.

Answer: We thank you for the recommendation., Line 120, we changed the title 2.2.

Q4: Table 1 – clinical signs of conjunctivititis before yes is 6 ?

Answer: In the revised version of the manuscript, we fixed this aspect (Line 177, table 1).

Q5: Conclusion: Suggesting about combination of staining and PCR techniques to diagnose is not appropriate. The confirmation of Chlamydia infection is based on PCR test. It is not necessary staning method before PCR which is time consuming and possible are false negative or doubtful. The PCR might be the only method for test. While staining method always must be confirmed by PCR.

Answer: The authors agree the reviewer opinion and rephrased the referred sentence resulting in (in the revised manuscript version, lines 306-309): "This study provides valuable information for veterinarians in managing cat ocular infections, suggesting that a combination of staining and PCR techniques can increase the accuracy of the diagnosis of the disease. However, the obtained results suggest that for the confirmation of Chlamydia infection the PCR technique is the best option, being recommended also in clarifying doubts during staining."

Q6: Moreover in 2.3 statistical analysis there is information about range of analysis but without agreement betweem screened methods while in discussion line 219-222 there is a sentence abiut results of this anlysis.

Answer: The authors regretfully omitted to complete the table with the mentioned territorial distribution analysis of the infection (in lines 219-222) within the metropolitan area. In the revised version, the authors completed the statistical analysis and Table 1 with this information (please see lines 39 and 211 of the revised version). Thank you for your understanding!

Reviewer 2 Report (Previous Reviewer 3)

The authors have responded to the reviewer's questions and revised the manuscript according to the reviewer's requests. After the correction of a few minor errors (see them below), the manuscript is suitable for publication and I support its publication.

Line 26: missing space Departmentof

Line 31: extra space after the work .

Line 146-147: The dot at the end of the sentence is missing. “Subsequently, we screened all of the conjunctival samples by PCR for molecular investigation”

Line 190-192: Please rephrase: Table 1 summarizes the results of Chlamydiaceae family detection using May Of the cats that tested positive by PCR, 72.6% (45/62) were asymptomatic and another 27.4% (17/62) expressed clinical signs of conjunctivitis –Grünwald–-Giemsa staining and the specific PCR method.

In the table, the decimal value of p values is sometimes indicated by a full stop/dot and sometimes by a comma. Please standardize.

Line 311: instead of increase use increasing

Line: 3500-352: Please rephrase: Stray cats are known to roam freely, and several animals may share a farm, which means that stray cats are possibly being fed in high-traffic areas, which may explain why stray cats have higher rates of chlamydial infections.

Author Response

Response Reviewer 2 (Revision 1)

The authors have responded to the reviewer's questions and revised the manuscript according to the reviewer's requests. After the correction of a few minor errors (see them below), the manuscript is suitable for publication and I support its publication.

Query (Q) 1: Line 26: missing space Departmentof

Answer: In the revised version of the manuscript, we fixed this aspect (Line 21).

Q2: Line 31: extra space after the work .

Answer: In the revised version of the manuscript, we fixed this aspect (Line 26).

Q3: Line 146-147: The dot at the end of the sentence is missing. “Subsequently, we screened all of the conjunctival samples by PCR for molecular investigation”

Answer: In the revised version of the manuscript, we fixed this aspect (Line 126).

Q4: Line 190-192: Please rephrase: Table 1 summarizes the results of Chlamydiaceae family detection using May Of the cats that tested positive by PCR, 72.6% (45/62) were asymptomatic and another 27.4% (17/62) expressed clinical signs of conjunctivitis –Grünwald–-Giemsa staining and the specific PCR method.

Answer: In the revised version of the manuscript, we corrected and rephrased this sentence (Lines 166-168).

Q5: In the table, the decimal value of p values is sometimes indicated by a full stop/dot and sometimes by a comma. Please standardize.

Answer: We thank you for the recommendation. In the revised version of the manuscript, I standardized this aspect and only used the dot (Line 177, table 1).

Q6: Line 311: instead of increase use increasing

Answer: In the revised version of the manuscript, we corrected this aspect (Line 320).

Q7: Line: 3500-352: Please rephrase: Stray cats are known to roam freely, and several animals may share a farm, which means that stray cats are possibly being fed in high-traffic areas, which may explain why stray cats have higher rates of chlamydial infections.

Answer:  We thank you for the recommendation. In the revised version of the manuscript, we corrected and rephrased this sentence (Lines 296-298) and replace it with this paragraph:  It is possible that the recorded higher prevalence of chlamydial infections in stray cats is due to the fact that they are fed in high-traffic areas and are free to roam, both situations increasing the likelihood of their exposure to other animals.

This manuscript is a resubmission of an earlier submission. The following is a list of the peer review reports and author responses from that submission.

Round 1

Reviewer 1 Report

Tirziu and colleagues investigated the occurrence of Chlamydia felis in stray cats in Timisora, Romania. Although the idea and intention of the study are interesting and justified on both animal welfare and public health grounds, data collection and especially analysis appear to be flawed and the present manuscript unsuitable for publication. Indeed, it is hard to follow the story line and not to get lost in the many inaccurate, misleading and sometimes contradictory information given.

The sampling procedure is well described (although ethical approval or comments are missing), but further processing of samples including cytological and PCR analysis and results thereof are not.

At one point it is said “Subsequently, all of the samples showing positive results for chlamydia infection were molecularly screened” (L96). However, later on it turned out that obviously all 95 samples were tested by PCR. A clear distinction or comparison of cytological (staining) and molecular (PCR) is missing. For example, in the abstract the authors just say: “65.3% (62/95) of the tested cats expressed positive results for C. felis”, but do not mention which method the statement is related to (L33).

Also, the following contradictory statements are completely puzzling and confuses the overall message of the manuscript: 1) L118: “All stray cats (27/95; 28.5%) with clinically suspected feline chlamydophilosis conjunctivitis were found to have C. felis.” But 2) L135: “C. felis was detected in 17/27 symptomatic (62.96%) … stray cats”. Probably this contradiction arises from the different diagnostic tools applied, but they are neither mentioned in a comprehensible way, nor is this difference discussed in the manuscript.

Further, result presentation as in the table does not make sense. Calculated percentages do not consider the total numbers of femals, males, age groups and so on. Therefore, in my opinion, the given percentages have no or little meaning. The authors claimed in M&M to calculate (or estimate???) a relative risk factor, but do not show how and do not present these data.

Further, the manuscript is badly structured (L222-225, L226-L231 and L232-L237 have nothing to do with risk factors as stated in the subheading). It is also full of errors (L118 and elsewhere “chlamydophiosis” is outdated for chlamydiosis) and redundancies (L239-241 redundant with L212-213, L238-239 with L216-217).

My recommendation is to completely re-structure data analysis and manuscript before re-submission. Concerning difficulties with proper epidemiological data acquisition and interpretation as well as with coherent and comprehensible descriptions, the involvement of an epidemiologist and a language assistant is highly recommended.  

Author Response

Dear reviewer, thanks for taking the time to review our manuscript and for your close attention to detail. We highly appreciate your corrections, suggestions, and comments, which significantly improved the quality of the submission. During the revision process, we tried to do our best to address each of these successfully. Please see below our responses in a point-by-point fashion to the raised concerns.

To be easily findable in the revised manuscript we marked all our answers/corrections in blue.

With due respect for your hard work and expertise,

Lecturer DVM Ph.D., M.Sc. DEGI Janos

Reviewer 2 Report

The study submitted by Tîrziu et al. investigated the occurrence of Chlamydia felis in asymptomatic and symptomatic stray cats in Timisora City, Western Romania, using staining and PCR methods. The study detected an overall C. felis PCR positivity in 62 out of 95 cats (65.3%).

This manuscript is a welcome contribution to studies from other countries regarding the C. felis prevalence in stray cats and uses both traditional staining methods and PCR for Chlamydia detection.

This paper is worth consideration, however there are a number of moderate to major and some minor concerns that need to be addressed.

In general, while the paper is well-written, some additional editing should be performed to avoid the use of uncommon phrases. Some suggestions were made in the list below.

Moreover, details in the Material and Methods section are lacking. A non-specific Chlamydia (formerly called Chlamydophila) PCR was used for C. felis identification. That, taken together with the non-specific staining methods only allows the identification of Chlamydia sp., not C. felis. These concerns must either be addressed (demonstration of species-specificity by addition of references or the appropriate data) or the manuscript changed accordingly (changing C. felis to Chlamydia sp. throughout the manuscript with appropriate changes in the introduction and the discussion).

Additionally, while the choices of statistical tests appear to be correct, the numbers in the table do not always match the numbers in the abstract or the text. For example, in the abstract and the Results section, it appears to be 2-fold more likely for asymptomatic cats to be positive than symptomatic cats. However, the table clearly shows that the number of overall symptomatic stray cat is lower (27 symptomatic cats, of which 17 were positive; 68 asymptomatic cats, of which 45 were positive) resulting in approximately the same proportion of PCR-positive cats per group (63% vs 66%). This section must be majorly revised before the manuscript is suitable for publication.

Finally, it was very interesting to read about Giemsa staining of smears to identify C. felis. While these staining methods are common in diagnostics (especially abortion diagnostics, C. abortus in small ruminants), species- or genus-specific antibody detection by immunofluorescence is far more common in current research papers. For C. felis, the method is rarely used. Therefore, it would be of great interest and benefit for the Chlamydia research community to see what Giemsa-stained chlamydial inclusions look like, and what exact protocol was used for staining. An addition of a figure with photos of Chlamydia-negative and positive stains is strongly recommended. Moreover, an elaboration on this topic in all the main sections of the manuscript (Introduction, Material and Methods, Results, Discussion) including a comparison between the sensitivity of the staining method and the PCR would strongly improve the quality of the manuscript. For example, it would be of interest if the lower sensitivity of the Giemsa stain is specific for one of the factors analyzed in this study (e.g. asymptomatic vs. symptomatic). This data could be supplied by a supplementary file and elaborated on in the text.

Minor comments and suggestions (by lines in the manuscript):

Title:

The title does not adequately reflect the data presented in the study. No investigation regarding the zoonotic potential of C. felis took place. Moreover, the investigated region should be included in the title.

Abstract:

Giemsa staining of smears was performed but is not mentioned in the abstract. The lower sensitivity of the smear (45/96 Giemsa, 62/95 PCR) is of interest for the Chlamydia research community

·       Line 29: Please abbreviate: Chlamydia (C.) felis

·       Line 29: “conventional and molecular tools” Please specify the methods (i.e. staining and PCR methods). PCR has become a conventional method to identify Chlamydia (see OIE manual)

·       Line 33: Please rephrase. For example, “There was no significant difference between…”

·       Line 34/35: Please rephrase. For example, “However, the isolation frequency of C. felis was significantly higher in asymptomatic compared to symptomatic cats (p=0.005).”

Introduction:

·       Line 49: Please rephrase. For example, change “information…is permanently required” to “more information… is needed”

·       Line 55-56: Perhaps adding “non-human” host would be more accurate concerning the examples given. Or referring to wildlife rather than potential hosts.

·       Line 55-56: There is no reference for this statement.

·       Line: 57: This is the first reference to Chlamydia felis in the main text (excluding abstract). Please, write Chlamydia (C.) felis to avoid confusion.

·       Line 57: Please rephrase. For example “intracellular pathogen with zoonotic potential”

·       Line 57 ff.: Cats are the main host of this species and may only be found in dogs and humans. The first reference provided (14) does not support the statement of C. felis occurring in birds. This section should be rephrased. Currently, this section does not adequately represent the frequency with which the mentioned hosts are infected with C. felis (cats before any other species), and does not provide information whether infection with C. felis is common in humans and what kind of symptoms (humans) and clinical signs (animals) can be expected. Moreover, these sentences starting at line 57f. appear to refer to the Chlamydia genus in general (host range, clinical signs), but only C. felis is mentioned. Please revise this section.

·       Line 60-61: There is no reference for this statement.

·       Line 60: What do the authors mean by ‘contagious’?

·       Line 62: Please rephrase: e.g. ‘and may be a nasal epithelium pathogen’

·       Line 63-64: There is no reference for this statement.

·       Line 64: The course of the disease…

·       Line 66: The reference format is incorrect.

Material and Methods:

In general, details for the methods are lacking. More details for both the staining method and the PCR method are necessary.

Moreover, the term ‘isolation frequency’ was used throughout the manuscript. However, the methods and results do not refer to any isolation techniques. Please, change the term ‘isolation frequency’.

·       Line 77 ff. It is not entirely clear whether one or two swabs were taken (one for smear and one for PCR, or one swab for both procedures). Please clarify.

·       Line 77: 'scattered in’ is an uncommon phrase for the distribution of stray cats in an area. Please rephrase.

·       Line 77: The last word is bold.

·       Line 88: There appear to be two spaces between ‘including’ and ‘age’

·       Line 90: Like in the abstract “conventional and molecular methods” is not an appropriate statement. Molecular methods (PCR) are ‘conventional’ in order to identify any Chlamydia species.

·       Line 93: Please correct to “in a first step”

·       Line 93 ff. Please briefly describe the staining protocol and mention the material used for the stain (smears on glass slides fixed in methanol described in the section above). Also, use the term ‘chlamydial inclusions’ rather than ‘inclusion bodies’

·       Line 96: “molecularly screen” is not a common phrase. Please rephrase to ‘screened by PCR’

·       Line 96ff: This PCR is not Chlamydia felis-specific according to the reference provided (Quote from the publication: “PCR detection of Chlamydophila spp was performed with primers designed by Buxton et al. (1996), modified by Raso et.al (2006), which corresponded to the conserved regions in the upstream non-coding region and 5' coding region of the chlamydial major outer membrane protein gene”). Has this PCR ever been validated in other publications?

The authors must confirm species-specificity of their results either by providing an appropriate reference or additional typing data to state that the Chlamydia found in these cats is Chlamydia felis (e.g. sequencing or species-specific PCR). Alternatively, the authors could adjust their manuscript to refer only to the detection of Chlamydia sp. rather than Chlamydia felis.

·       Line 97: ‘detect’ instead of ‘evidence’

·       Line 101: C. felis is not in italics

·       Line 102: put ‘PCR grade distilled water’ within parentheses and delete the word ‘meaning’

·       Section ‘2.3 Statistics’: What tests were used to determine the p-values? What program was used for statistical analysis? Please provide more detailed information

Results:

·       Line 114-115: This sentence needs revision: “Overall, 45 (47.4%) out of 95 specimens' cytological results revealed basophilic, oval intracytoplasmic inclusion of ocular epithelial cells using May-Grünwald-Giemsa staining.” Specifically, the term ‘cytological’ was not used in the Material and Methods section. Please use the term ‘smear’ or ‘cytological’ or a similar term consistently throughout the manuscript. “Inclusions of ocular epithelial cells” is not the correct terminology. Were magenta-stained inclusions found in ocular epithelial cells? A figure with chlamydial inclusions from the smears would be of great interest to the research community.

·       Line 116f: The authors mention 17 smears that were negative in the Giemsa-stain, probably referring to the 17 additional samples that were positive in the PCR but not the smear. However, the current text does not explain why only 17 negative smears rather than 50 negative smears were mentioned. Please mention the PCR results earlier in the text or mention all 50 negative smears.

·       Line 128/129: How was contact to humans assessed in this study?

·       Line 135/136: ‘totaling 62/95 (65.3%) positive cats’ Please adjust to ‘resulting in … PCR-positive cats’

·       Line 139/140: These sentences are incomplete. Please use full sentences for this section.

·       Line 142: Use % instead of ‘per cent’ as was applied above; please adjust throughout the text

·       Line 142: ‘confirmed PCR-positive’… Since all statistical data was performed with the PCR results, please specify this throughout the text. A statistical comparison between staining and PCR sensitivity would be of great interest to the research community, however.

·       Line 143-145: How is this data represented by Table 1? The total number of positive symptomatic cats (62.96%) was similar to that of asymptomatic cats (66.17%)

·       Line 145: Use lowercase p for p-value as done above

·       Line 149: Use ‘sex’ instead of ‘gender’

·       Line 150: Typographical error: change ‘famele’ to ‘female’

·       Line 151/152: Was this difference statistically significant? No p-value has been provided.

·       Line 153: Did the authors intend to make a reference to disease or to infection?

Discussion

·       Please correct ‘Discussions’ to ‘Discussion’

·       Line 159: Please, put C. felis in italics

·       Line 161: The use of apostrophes is uncommon in the English language. Please write ‘upstream non-coding region of the…”

·       Lines 160-162: Please add references

·       Lines 163f: Please add references where applicable. Moreover, the first statement is not entirely supported by the data presented given that both asymptomatic and symptomatic cats were positive for C. felis. Please adjust accordingly.

·       Line 170: Please write ‘According to the people who rescued’. This should be mentioned in the Results section and discussed in this section or entirely omitted because it was not the focus of the study.

·       Line 170-172: This belongs into the Results section.

·       Line 173-176: Please add references for all statements mentioned here. Moreover, this would belong into the introduction section as it does not discuss the results of the study.

·       Line 177: Were isolation or staining methods used for detection? Please clarify.

·       Lines 179-180: Write % instead of ‘per cent’

·       Line 182-184: Please provide references for these statements.

·       Line 185: What was not followed up on? I would either omit this sentence or clearly state the limitations of your study.

·       Line 189: The zoonotic potential of C. felis has been presented in other study. Please discuss and provide references

·       Line 193: Write % instead of ‘per cent’

·       Line 197: Write % instead of ‘per cent’

·       Line 200: Please add a reference

·       Line 202: ‘detect’ instead of ‘capture’

·       Line 203: Please rephrase. ‘…, the prevalence of C. felis in cats remains unclear.’

·       Line 212: ‘The association of cat age with C. felis infection is more common in cats younger than one year old’ Unusual phrasing. Please rephrase.

·       Line 215, 216, 218, 220 (and more; please search for the term and replace it): Chlamydophila is an outdated term. Therefore, use the word ‘chlamydiosis’ rather than ‘chlamydophilosis’.

·       Line 219: Was there a follow-up? What co-infections were assumed? Please elaborate.

·       Line 220: Have any other studies found these clinical signs?

·       Line 223: Please add a reference for this first statement. It is unclear which of the references in line 225 refer to this statement.

·       Line 226-227: Please add references

·       Line 228: Here, the authors speak of ‘relapsing’ infection but no repeated sampling was performed. Please, clarify this statement.

·       Line 229: Please add references for this statement. In most studies, C. felis infections only result in conjunctivitis

·       Line 234: ‘essential’ does not appear appropriate in this context. Please rephrase.

·       Line 237: Please add the missing parentheses

·       Line 241: This has already been stated previously

Conclusion

·       Lines 245-248: This part of the conclusion is partially incorrect. Isolation was never a major focus for diagnostics and molecular diagnostics (PCR) for the detection of C. felis and other veterinary Chlamydia species have long been established. However, it is true that the gradual replacement of staining methods with PCR methods is recent. Another flawed conclusion is the concern for mutations. As Laroucau et al. 2012 has clearly shown, C. felis is a highly conserved chlamydial species with a very low mutation rate. Concerns for potential evasion from detection by mutation is therefore highly unlikely. Please revise this section. Also, for the staining/molecular diagnostics comparison, the two methods need to be better compared in the Results and discussion section. Otherwise, the first part of the conclusion does not properly reflect the nature of the study

Author Response

(The authors gave the same response as above.)

Reviewer 3 Report

The topic of this article is to investigate the prevalence of Chlamydia felis in healthy and conjunctivitised cats in Romania. The authors support the findings of previous international studies.

Major points:

1.       In several places in the article, the author uses the term chlamydophilosis for the disease caused by Chlamydia felis, which is no longer appropriate. The genus Chlamydophila has been discontinued and the term Chlamydia felis is now used, so the disease it causes is now called chlamydiosis.

2.       The author claims that a Chlamydiaceae-specific primer was used to detect Chlamydia felis. My question is what are the chances that Chlamydia pneumoniae was the causative agent and not Chlamydia felis? Literature suggests that Chlamydia pneumoniae can also cause conjunctivitis in cats.

3.       (Detection of Chlamydophila pneumoniae in cats with conjunctivitis. Sibitz C, Rudnay EC, Wabnegger L, Spergser J, Apfalter P, Nell B. Vet Ophthalmol. 2011 Sep;14 Suppl 1:67-74. doi: 10.1111/j.1463-5224.2011.00919.x. PMID: 21923826)

4.       In the Abstract it can be read. "Conjunctival samples were harvested from 68 clinically healthy cats and another 27 cats presenting clinical signs of conjunctivitis".  In the Materials and Methods section, it can be read the opposite: The assessment of the presence (n=68) or absence (n=27) of conjunctivitis was carried out by a veterinarian who was a specialized ophthalmologist. What is the truth?

5.       I suggest that since the extracted nucleic acids are available, PCR testing with primers specific for Chlamydia felis should also be performed. If this is not done, based on the results available, they can only say that x% of stray cats in Romania was found to be Chlamydia positive.

6.       The description of the methods is quite incomplete. The description does not allow the methods used to be replicated. I suggest that the exact procedure of nucleic acid extraction, PCR, and nucleic acid detection should be described with the name of the reagents used and their origin.

7.       I would advise the authors to edit the article, especially the table, which is a bit monotonous and try to present the results in figures.

8.       It might be a good addition to the article and would raise its quality, to show some smears from eye specimens painted with the May-Grünwald-Giemsa technique in pictures.

9.       The manuscript should be supplemented with relevant references in several places.

Minor point:

In the Abstract, line 31 the number is incorrect

In the Abstract, lines 34 and 36 italicize the P

In the Abstract, line change sourses for sources

In line 58: “This pathogen infects humans.” reliable reference is needed

In line 61:a  reference is needed

In line 66: the incorrect style of the reference (Sykes, Gruffydd), please use a number

In line 71: consider periurban, which was written earlier in line 43 as peri-urban

In line 77: should not be bolded “the”

In line 85: please cancel Waltham, Massachusetts, USA.  The name of the town, state, and country is not needed after the first mention.

In line 96: replace polymerasic for polymerase

In lines 110 and 135: italicize the P

In line 137: So far, the author of the article has used lower case p as a sign of significance, not capital P. Italicize also it.

In line 176: a reference is needed.

In line 228: The author has written: “This study demonstrates the occurrence of relapsing C. felis infections in cats living in high-population areas.” What do you mean by  “relapsing”?

In line 236 change populace for population

In line 237 I think the author should cancel the next from the text: Thermo Fisher Scientific, Waltham, Massachusetts, USA

Author Response

(The authors gave the same response as above.)

Author Response

(The authors gave the same response as above.)

Round 2

Reviewer 1 Report

Unfortunately, the manuscript has not been significantly improved. The point-by-point answers are very general and most of the issues have not been addressed. The standard answer for most reviewer's comments was "We redid Table 1, correcting the data throughout the revised manuscript (please see the submitted manuscript)." However, corrections were not made or are insufficient. Therefore, almost all critical issues from my first review still apply. The data and analysis shown in the table are unclear and difficult to understand, e.g. it is not clear what "overall" means and how the numbers were calculated.

Besides the fact that used methods are rather outdated (an unspecific conventional PCR and staining with low sensitivity), the greatest weakness of the study is the poor and erratic data analysis and interpretation. One example is already in the abstract L32-35. It is meaningless to state that 72 % of PCR-positives were assymptomatic and 27 % symptomatic because it does not answer the question whether there is an association between the presence of chlamydiae and disease. There are many more examples for similar meaningless statements throughout the manuscript.

Further, the language is still poor and contains errors. In my eyes, the manuscript is not worthy of publication.

Reviewer 2 Report

I thank the authors for their corrections and excellent point-by-point response. One important and few minor points remain that should be addressed prior to acceptance of the manuscript:

Specifically, in your statistical analysis, has the sampling bias of symptomatic vs. asymptomatic cats been properly considered? Given the data presented, I am not convinced that asymptomatic cats are more likely to be Chlamydia-positive than symptomatic cats given that much more asymptomatic cats were sampled. See comments below (line 238). Please elaborate in the response how the analysis was conducted and how the sampling bias was taken into consideration.

Additional comments and suggestions:

Line 105: Here is a typo with "I5" instead of 15. Please correct.

Table 1: Typo in clinical signs of conjunctivitis “6Yes”. Please correct. Additional, there is an unnecessary line between 9 and 69.2% and 8 and 57.1%.

Line 149-151: Does this information refer to Table 1? If yes, please specify in parentheses (Table 1).

Line 175: The semicolons make the sentences difficult to read. Throughout the manuscripts, the reader might appreciate the following format: e.g. “17/27 (62.9%)” or “62.9% (17/27)” instead of “17/27; 62.9%”

Line 191: Given the individual differences (Giemsa-positive but PCR-negative and vice versa), which are not clearly shown in table 1, a supplementary table (Excel or csv file) with all individual samples and their Giemsa/PCR results would be of interest and would complete the data set presented. Moreover, please note the exact number of Giemsa-neg/PCR-pos samples and vice versa in the main text.

Line 199: Please correct: Isolation ‘in’ cell culture.

Line 219-220: This is a very important observation. Perhaps, to further stress out this point, you could rephrase these lines as follows:

While we detected a fair agreement (k index = 0.308) between the two screening methods used (PCR, staining method), the occurrence of PCR-positive/Giemsa-negative (n=X samples) and PCR-negative/Giemsa-positive (n=X) samples suggest that the combined use of both screening methods is recommended to properly evaluate the infection status of cats.”

Line 224: Please correct to ‘analyses’

Line 238ff: There is still the sampling bias to consider: Twice as many asymptomatic cats were sampled compared to cats with conjunctivitis. When statistical analysis was performed, was this bias taken into consideration?

Line 254f. Please move this to line 233 where the age data is presented.

Line 290: Please correct to “(…) for veterinarians to manage ocular infections in cats, suggesting (…)”

Line 293: You could also add under limitations of the study that the Chlamydia species has yet to be identified, although C. felis is the most likely candidate.

Reviewer 3 Report

Based on the reviewers' questions and comments, the authors have improved the manuscript, which is suitable for publication in this form.

Reviewer 4 Report

Authors tried improved manuscript according with reviewers suggestion but still the manuscript is enought clear. PCR method used by authors are not sufficient they should identified all Chlamydia species. The PCRs for identification other than C. felis species are available so if there are postive samples for Chlamydia spp. authors should identified species with using specyfic PCRs. Morever control of inhibition is absent in PCR.